# Visible-Light-Active Vanadium and Copper Co-Doped gCN Nanosheets with Double Direct Z-Scheme Heterojunctions for Photocatalytic Removal of Monocrotophos Pesticide in Water

Dhanapal Vasu [1,2], Arjunan Karthi Keyan [1,2], Subramanian Sakthinathan [1,2], Chung-Lun Yu [1,2], Yu-Feng You [1,2], Te-Wei Chiu [1,2,*], Liangdong Fan [3,*] and Po-Chou Chen [4,5]

[1] Department of Materials and Mineral Resources Engineering, National Taipei University of Technology, No. 1, Section 3, Chung-Hsiao East Road, Taipei 106, Taiwan
[2] Institute of Materials Science and Engineering, National Taipei University of Technology, No. 1, Section 3, Chung-Hsiao East Road, Taipei 106, Taiwan
[3] Department of New Energy Science and Technology, College of Chemistry and Environmental Engineering, Shenzhen University, 3688 Nanhai Ave., Shenzhen 518060, China
[4] Graduate Institute of Organic and Polymeric Materials, National Taipei University of Technology, Taipei 10553, Taiwan
[5] E-Current Co. Ltd., 10F.-5, No. 50, Sec. 4, Nanjing E. Rd., Songshan Dist., Taipei 10553, Taiwan
* Correspondence: tewei@ntut.edu.tw (T.-W.C.); fanld@szu.edu.cn (L.F.)

**Abstract:** In this study, both vanadium and copper elements were anchored on graphitic carbon nitride (gCN) (denoted as V/Cu/gCN) via a thermal decomposition process as a novel nanosheet photocatalyst for the removal of monocrotophos (MCP). The prepared nanosheet features were studied by utilizing XRD, UV–Visible absorption spectrometry, PL, FE-SEM, TEM, and XPS techniques. These analytical techniques revealed the successful formation of direct Z-scheme heterojunctions of V/Cu/gCN nanosheets. The dopant materials significantly enhanced the electron–hole separation and enhanced the removal rate of MCP as compared with bulk gCN. The investigation of effective operating conditions confirmed that a higher removal of MCP could be obtained at a doping concentration of 0.3 wt% and a catalytic dosage of 8 mg with 80 min of visible-light irradiation. The generation of various reactive radicals during the degradation process of the photocatalyst was observed using a scavenging treatment process. Additionally, the scavenging process confirmed that $e^-$, OH•, $h^+$, and $O_2^{•-}$ played a major role in MCP degradation. The direct Z-scheme dual-heterojunction mechanism, as well as the possible pathway for the fragmentation of MCP by the V/Cu/gCN nanosheet photocatalyst, was derived in detail. This research article provides a novel perspective on the formation of excellent semiconductor photocatalysts, which exhibit enormous potential for environmental treatments.

**Keywords:** vanadium; gCN; copper; photocatalysis; AOP; nanosheets; MCP

## 1. Introduction

The growing environmental pollution levels of harmful chemicals, organic molecules, dyes, heavy metals, pharmaceutical drugs, and pesticides have been majorly responsible for contaminating the global atmosphere [1]. Among the many pollutants, agricultural pesticides used for the purpose of protecting plants are the most harmful and are quite hazardous to human health. Pesticides are most commonly used in agricultural applications to control and prevent the growth of pests and to enhance crop production, whereas their heavy utilization can be harmful to water resources and soil [2–4]. High concentrations of pesticides in crops cause high contamination in the atmosphere and are ultimately toxic to humans. Therefore, for any agricultural purpose, limited quantities of insecticides and pesticides contribute to plant growth, while the remaining 20% can be absorbed onto the soil surface, and insoluble compounds can eventually mix with water resources, resulting

in toxicity to food, water, and soil [4–6]. Sraw et al. [1,2] reported that pesticides are the second most common pollutants present in water resources. It has been mentioned that a wide variety of these pesticides cause mutations in DNA cells, resulting in the formation of cancer and damaging the endocrine and nervous systems of the human body, even at low concentrations. Monocrotophos (MCP), a type of organophosphorus pesticide, is commonly used for protecting cotton plants from insects [2,6–8]. MCP is generally utilized to serve as a phosphorous source for plant growth. The hydrophilic behavior of MCP gives it better solubility in water (1 kg/kg, 20 °C). Generally, MCP molecules easily bond to water molecules because they easily travel with water downstream to the soil. MCP and its derivative's half-life mainly depends on the pH and temperature and can be approximately estimated at 17–96 days. Therefore, the World Health Organization (WHO) has characterized this pesticide in the class I category due to its highly toxic nature [2,8–10]. Therefore, the degradation of this pollutant in water resources has become a very challenging and critical concern drawing the attention of the research community in recent decades.

To rectify these issues, a lot of techniques or methods have been adopted, such as nanofiltration, adsorption, electrocoagulation, biological degradation, and advanced oxidation processes, to eliminate or remove pollutants from water resources [1,2]. A wide variety of advanced oxidation techniques and related materials have been produced, including Fenton/Fenton-based reactions, photocatalysis, electrochemical oxidation, and ultraviolet (UV) irradiation [6–9]. Among these removal techniques, semiconductor photocatalytic advanced oxidation processes (AOPs) have been widely used as a fitting method for the removal of MCP in water resources [8–10]. It is most important to develop an efficient and suitable photocatalyst with sustainability and abundance. Among photocatalysts, 2D-layered graphitic carbon nitride (gCN) has gained more attention for its structure, chemical stability, non-toxic nature, and wide bandgap (Eg = 2.7 eV) [11–13]. Nevertheless, the physio-chemical properties of bulk gCN have some limitations, such as activity under UV light, low surface area, fewer active sites, and rapid electron–hole pair recombination [14,15]. To solve these challenges and enhance the photocatalytic behavior of bulk gCN, lots of effective methods have been utilized, such as element doping, morphological modifications, and heterojunction construction [16–18]. Several charge-transfer reactions are considered to enhance the interfacial effect of transition metal oxide semiconductors and gCN upon coupling [12–18]. In some reports, the generation of direct solid Z-scheme heterojunctions has been derived. Direct Z-scheme double-heterojunction nanosheets can enhance the activity of individual oxides. In addition, these heterojunctions increase and restrict photoinduced electron–hole pair formation and recombination. Therefore, the charge-carrier lifetime increases with higher redox potentials, resulting in higher photocatalytic efficiency [19]. Natural photosynthetic systems have been utilized to discover a mechanism that provides the possibility of using two narrow-bandgap semiconductor materials with many active sites and photoinduced electron–hole pairs [11–13]. Due to this outstanding characteristic, it can rectify the shortcoming of the p-n heterojunction generated between V and Cu. To the best of our knowledge, there are no reports available on the doping of vanadium and copper into the gCN lattice for the photocatalytic removal of organic pollutants.

This study aimed to prepare a direct Z-scheme for two-heterojunction nanosheets based on the anchoring of V and Cu elements onto the gCN host lattice, as well as to study the performance of the prepared nanosheets in the visible-light-driven photocatalytic mineralization of MCP. The effect of various parameters on the mineralization rate of MCP was investigated. Moreover, a possible degradation mechanism and mechanistic pathway for the mineralization of MCP were proposed.

## 2. Results and Discussion

### 2.1. Crystallographic Studies

The crystalline patterns of as-prepared (V/Cu/gCN) nanosheets are shown in Figure 1. The characteristic peaks corresponding to gCN can be identified. The two main characteristic peaks are observed at 2θ of 27.3 and 12.7, which correspond to the (002) and (100) planes of gCN, respectively [11–15]. The main high-intensity peak is present due to the interplanar stacking structures of graphitic carbon nitride. The other weak diffraction peak corresponds to the in-plane structural packing of gCN [19–23]. Undoped gCN compared with the V/Cu-doped gCN nanosheets underwent a significant shift of (002) lines, indicating that the gCN stacking of V/Cu was affected by the incorporation of V and Cu species in the gCN host lattice [23–27]. Moreover, the peak intensity of (002) was drastically decreased when the doping concentrations of the materials V and Cu increased [28,29]. In addition, the XRD crystalline structure revealed that when the doping concentration varied, the diffraction patterns also changed. The prepared nanosheet's crystalline size, full-width half-maximum (FWHM), and micro-strain [21] were also calculated, which are presented in Table 1. The prepared samples' crystalline size was obtained by using the Debye–Scherrer (D = kλ/β cos θ) equation. Furthermore, the prepared materials' micro-strain (ε) was observed by using the Williamson–Hall method (ε = B/4 tan θ). In addition, the metallic and oxide phases of V and Cu were not observed in the XRD diffraction patterns. Moreover, all V/Cu/gCN nanosheet diffraction peaks were slightly shifted and still had much weaker intensities, which revealed the successful incorporation of V/Cu/gCN nanosheets.

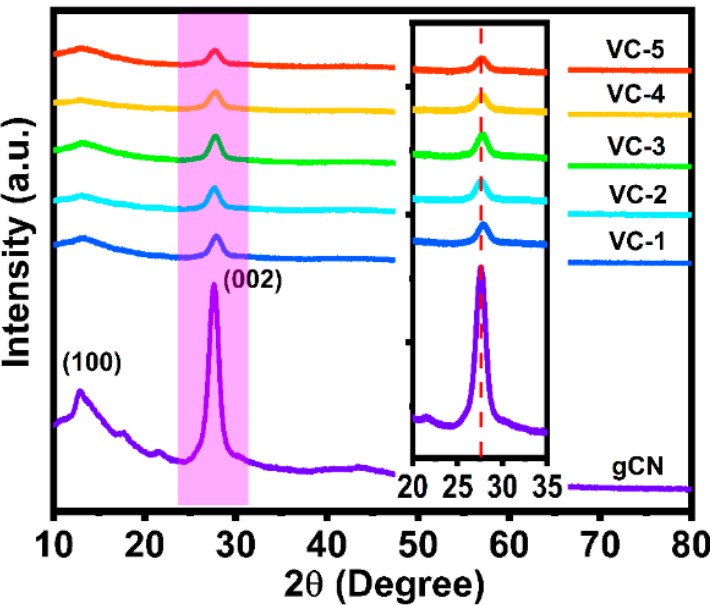

**Figure 1.** XRD studies for undoped and doped (V and Cu) gCN nanosheets.

**Table 1.** The prepared samples' FWHM, micro-strain, bandgap, and crystalline size.

| Sample ID | FWHM | Micro-Strain | Crystalline Size | Bandgap (eV) |
|---|---|---|---|---|
| gCN | 1.437 | 0.0255 | 5.95 | 2.19 |
| VC-1 | 1.71603 | 0.0304 | 4.98 | 2.12 |
| VC-2 | 1.6712 | 0.0296 | 5.12 | 1.93 |
| VC-3 | 1.94315 | 0.0345 | 4.4 | 1.51 |
| VC-4 | 1.83126 | 0.0325 | 4.67 | 1.96 |
| VC-5 | 1.72968 | 0.0307 | 4.94 | 2.1 |

### 2.2. Study of Absorption and Optical Properties

The prepared nanosheets' optical properties, e.g., bandgap, were observed by using UV–Visible absorption. Undoped gCN exhibited typical semiconductor absorption in the UV regions, originating from the gCN VB charge-transfer response populated by N 2p orbitals to the LUMO formed by C 2p orbitals [19–21]. The absorption peaks correspond to the aromatic rings' $\pi \rightarrow \pi^*$ transition, and the $n \rightarrow \pi^*$ transition is caused by an electron transfer from a non-bonding orbital to an anti-bonding aromatic group [30,31]. The bulk gCN bandgap is estimated to be 2.7 eV, according to the literature results. After the incorporation of V and Cu into the gCN host lattice, the absorption peak intensities and absorption width significantly increased in the visible-light region. Notably, the absorption edges of the doped gCN nanosheets shifted to longer-wavelength regions compared to bulk gCN. The optical energy bandgaps (Egap) of all nanosheets were examined by extrapolating the Tauc plot using the following expression [31].

$$\alpha h\nu = A\,(h\nu - Eg)^{1/2} \tag{1}$$

where $\alpha$ denotes the coefficient of absorption, h is Planck's constant, $\nu$ is the frequency of light, A is a constant, and Eg is bandgap energy. The equivalent bandgap energies of the prepared gCN, VC-1, VC-2, VC-3, VC-4, and VC-5 are 2.19, 2.12, 1.93, 1.51, 1.96, and 2.10 (Figure 2a–f), respectively.

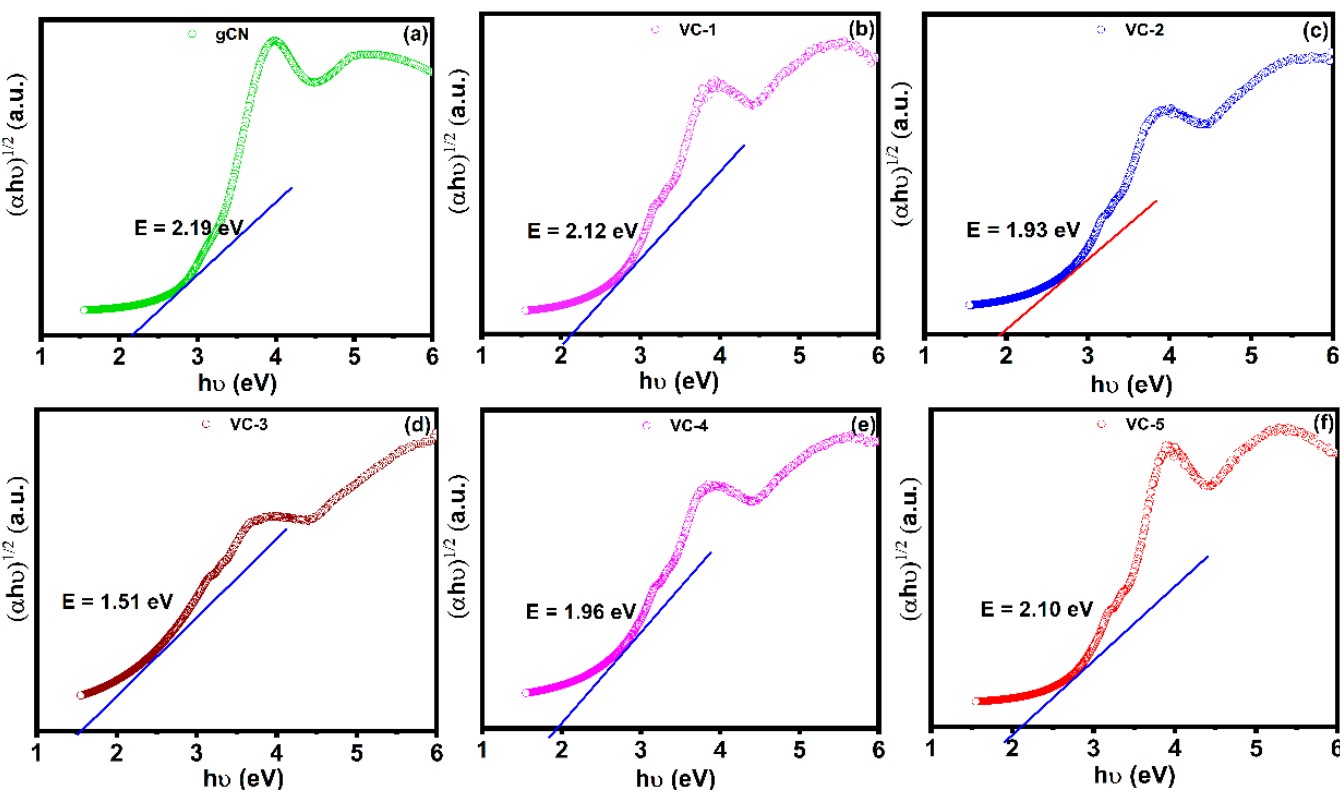

**Figure 2.** Tauc plots the bandgaps of all V/P/gCN nanosheets: (**a**) gCN, (**b**) VC-1, (**c**) VC-2, (**d**) VC-3, (**e**) VC-4, and (**f**) VC-5.

The energy bandgap (Eg) of VC-3 is much smaller than those of the other prepared nanosheets, indicating enhanced visible-light-trapping characteristics. Given that the hole oxidation property corresponds to the valence band position, V and Cu inclusion cause the valence band position to shift in the positive direction, and surface defects result in excellent photocatalytic activity [19,25,26,30]. The photogenerated charge carriers' splitting rates in the VC-3 nanosheet are higher, which is caused by photoinduced electrons in the defect sites, restricting the rapid recombination of the charge carriers. The results indicate that

V and Cu synergistically modulate the gCN host lattice and form a direct Z-scheme band structure resulting in a visible-light-active range and extraordinary visible-light utilization efficiency.

### 2.3. PL Study

The prepared gCN and doped gCN nanosheets' PL emission spectra are shown in Figure 3. It is confirmed that there was a drastic rise in photoluminescence quenching after the incorporation of V and Cu into the gCN host lattice. The PL emission peak at 440 nm for bulk gCN corresponds to the band transition, with the excitation energy greater than the bandgap energy [32]. The gCN LUMO or conduction band is generated by $sp^2$-hybridized clusters of C-N, whereas the HOMO or valence band is formed by nitrogen 2p orbitals, generally the lone pairs of 'N' in the tri-s-triazine structure [32–34]. The transition arises from C 2s2p and N 2s2p hybridization. Hence, dopant material hybridization with the gCN host lattice created midgap energy states below the CB of gCN, which significantly enhanced the absorption of photons, resulting in higher photocatalytic activity. With the incorporation of V and Cu into gCN, the emission intensity of PL was very low, which indicates that the incorporation led to an increased separation rate for photoinduced electron–hole pairs. Therefore, photoinduced electrons and holes will take a longer time to recombine [35]. After the incorporation of undoped gCN with V and Cu, a defect midgap state will possibly be generated just underneath the conduction band, which is at a lower energy level. In this process, the dopant defect site acts as an electron-trapping site. Therefore, conduction-band electrons will be scavenged in the V and Cu doping site instead of directly reaching the valence band of gCN. This process further leads to the reduced recombination of photoinduced electron–hole pairs, resulting in higher visible-light utilization. The V and Cu incorporation gives rise to dislocations in the gCN host lattice, which leads to enhanced surface energy.

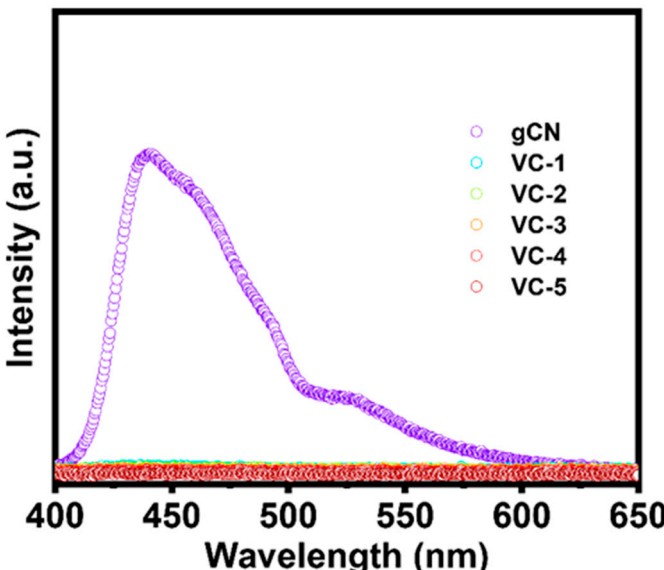

**Figure 3.** The PL emission spectra of gCN and V/Cu/gCN nanosheets.

### 2.4. Surface Morphology Studies

FE-SEM was utilized to study the surface morphologies of the prepared undoped gCN and V/Cu/gCN nanosheet samples. Figure 4 shows that the prepared materials have a highly porous nature with a layered structure (Figure S1), whereas no other metal aggregation is observed. In contrast, the FE-SEM images of all V/Cu/gCN materials, shown in Figure 4a–f, also show sheet-like surface morphologies [36]. These results suggest that the gCN host lattice successfully incorporated V and Cu heteroatoms.

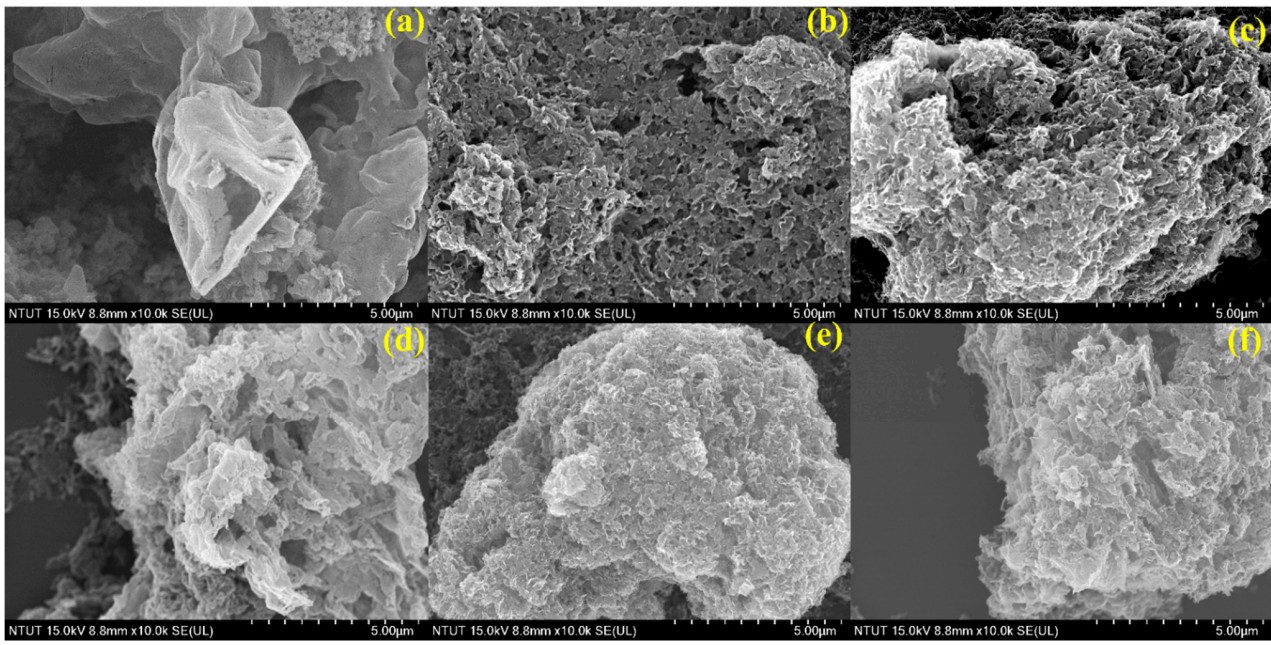

**Figure 4.** Surface morphology studies of bulk gCN and V/Cu/gCN nanosheets: (**a**) bulk gCN (**b**) VC-1, (**c**) VC-2, (**d**) VC-3, (**e**) VC-4, and (**f**) VC-5.

The prepared bulk gCN and doped gCN surface morphologies and the nanostructure of the nanosheets were studied using HR-TEM, which are shown in Figure 5. As shown in Figure 5a, bulk gCN possesses more wrinkles, and it is formed with thin sheet-like morphology [36]. From the HR-TEM images of doped gCN in Figure 5b, we confirmed the nanosheet-like structure of highly porous materials [20–22]. Moreover, according to the HR-TEM results, the dopant materials successfully formed a sheet-like structure, which is similar to the previously reported results on gCN systems. Indeed, the results indicate that pristine gCN nanosheets exhibit a wrinkled 2D lamellar structure combined with single layers. Hence, this unique structure leads to a higher surface area, which is suitable for an active catalyst.

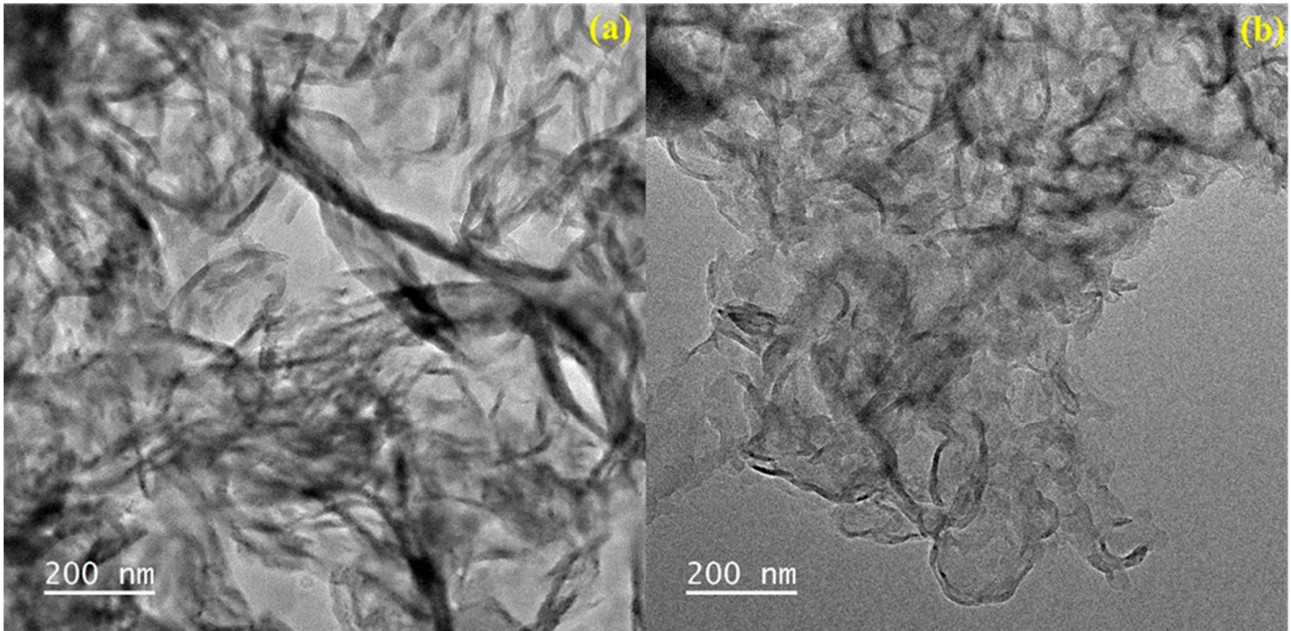

**Figure 5.** Surface topography studies of (**a**) bulk and (**b**) doped gCN nanosheets.

### 2.5. Surface Chemical Distribution Studies

The as-prepared bulk gCN and doped (V and Cu) gCN nanosheets' surface element distribution and the interaction between gCN and dopant materials were studied by XPS. In C1s, core-level spectra could be deconvoluted into three main peaks, 284.6, 286.07, and 287.86 eV, corresponding to those of gCN [11–14]. The adventitious C 1s carbon spectra were fixed at 284.6 eV due to the charge-referencing method. According to the core-level high-resolution spectra of C1s (Figure 6a), there is a small high-agreement peak observed at 287.86 eV, which corresponds to $sp^2$-bonded carbon (N-C-N) in the heptazine or triazine ring structure [14–18]. The small intense signal presented at 286.07 eV is assigned to the carbon atoms in the C-NH$_2$ groups. The binding energy signal at 284.6 eV is in agreement with graphitic carbon C-C coordination, which is commonly used for energy calibration. The binding energy of C1s at 287.86 eV for V/Cu/gCN nanosheets shifted to a significantly higher value (287.86 to 288.01) compared to that for undoped gCN, and the shifting corresponds to the interactions between gCN and dopant materials (V and Cu) [12,15,18]. In addition, the peaks related to the area of bulk gCN and doped gCN are given in Table 2. The high-resolution N1s core-level spectra could be deconvoluted into four fitted peaks at 398.34, 399.05, 399.92, and 400.81 eV. In addition, the carbon functional group's peak ratio varied between undoped and doped gCN samples, which indicated that $sp^2$ C was replaced by a nitrogen atom, resulting in the doping site being consistent with that of vanadium- and copper-doped gCN. These peaks could be assigned to the pyridine N, pyrrolic N, graphitic N, and quaternary C-N bond (see Figure 6b), respectively [12,15,18]. Generally, the N atoms are trigonally bonded to the three $sp^2$ carbon atoms in the gCN network (graphitic-like nitrogen structure). The pyridine N binding energy in the doped gCN sample increased by 0.1 eV compared with that of bulk gCN, also indicating the interfacial interaction between gCN and dopant materials (V and Cu). On the other hand, the C-N peak intensity decrease was ascribed to the substitution of the dopant materials, which is presented in Table 2. The presence of pyrrolic/graphitic nitrogen groups is more capable of producing more active sites for photocatalytic activity while providing a strong chemical interconnection between gCN and dopant materials for rapid electron transfer.

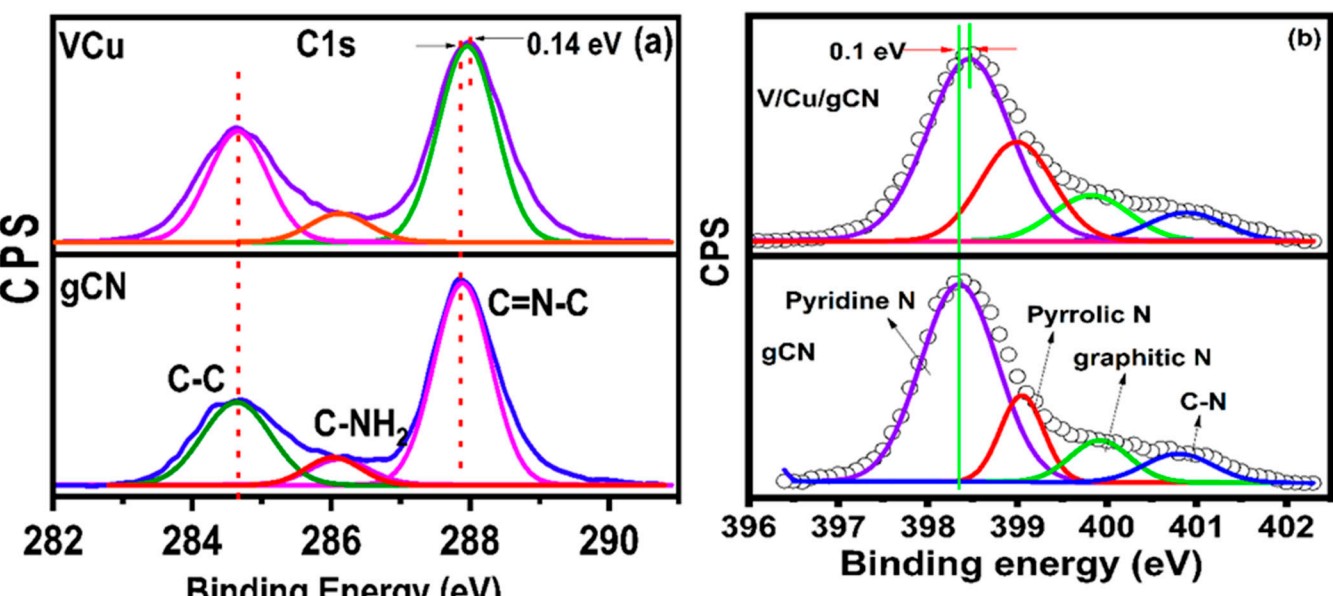

**Figure 6.** XPS spectra of nanosheets: (**a**) C1s spectra of gCN and V/Cu/gCN nanosheets and (**b**) N1s spectra of gCN and V/Cu/gCN nanosheets.

**Table 2.** C1s and N1s functional group distributions.

| | C1s | | | N1s | |
|---|---|---|---|---|---|
| Group | gCN (%) | V/Cu/gCN | Group | gCN (%) | V/Cu/gCN |
| C=N-C | 61.73 | 56.88 | Pyridine N | 64.97 | 55.26 |
| C-C | 30.19 | 34.22 | Pyrrolic N | 15.98 | 24.91 |
| C-NH$_2$ | 08.07 | 08.89 | Graphitic N | 10.62 | 12.22 |
| - | - | - | C-N | 08.41 | 07.58 |

The high-resolution XPS V 2p core-level spectra could be deconvoluted into six peaks at 513.94, 515.50, 516.46, 517.48, 523.06, and 524.21 eV (Figure 7a), revealing the oxidation states of V atoms in the V/Cu/gCN sample [19–21]. The high-resolution V 2p spectra reveal the formation of metallic $V^{2+}$ (VO), $V^{4+}$ (VO$_2$), and $V^{5+}$ (V$_2$O$_5$) with various binding energies. These overlapping characteristic spectra are commonly obtained in ternary vanadium oxide bronzes, which have different valence oxidation states of $V^{2+}$, $V^{4+}$, and $V^{5+}$. The high-resolution V 2p XPS spectra indicate the presence of three states [37,38]. VO$_2$ has very interesting electronic properties, and V$_2$O$_5$ is the common stable oxide species in vanadium due to its higher O/V ratio. The XPS fitting spectra indicate that $V^{5+}$ is the most dominant component in doped gCN, the ratio of which is higher than that of $V^{4+}$, as indicated by the V 2p$_{3/2}$ and V 2p$_{1/2}$ peaks. The higher concentration of $V^{5+}$ was the source of the V precursor, whereas the $V^{2+}$ and $V^{4+}$ oxidation states might have been generated during thermal decomposition by the reduction of V5+ species. The different vanadium oxidation states of $V^{2+}$, $V^{4+}$, and $V^{5+}$ tend to be on the gCN lattice surface, enhancing the separation efficiency of photoinduced charge carriers [39,40]. The dopant Cu element's chemical states with different oxidation states were observed after high-resolution core-level XPS deconvolution. Figure 7b shows that the deconvoluted Cu core-level Cu 2p XPS spectrum presents binding energies at 932.43 eV and 952.18 eV, which are attributed to the Cu$^+$ oxidation states of Cu 2p$_{3/2}$ and Cu 2p$_{1/2}$, respectively [41–43]. Likewise, the binding energies at 935.47 eV and 956.85 eV are related to the $Cu^{2+}$ oxidation states of Cu 2p$_{3/2}$ and Cu 2p$_{1/2}$. Moreover, minor satellite peaks related to $Cu^{2+}$ oxidation states also indicate binding energies at 942.4 and 938.08 eV [44–47], as shown in Figure 7b. Therefore, as per the Cu core-level XPS spectra, both CuO and Cu$_2$O exist on the gCN surface of the nanosheets.

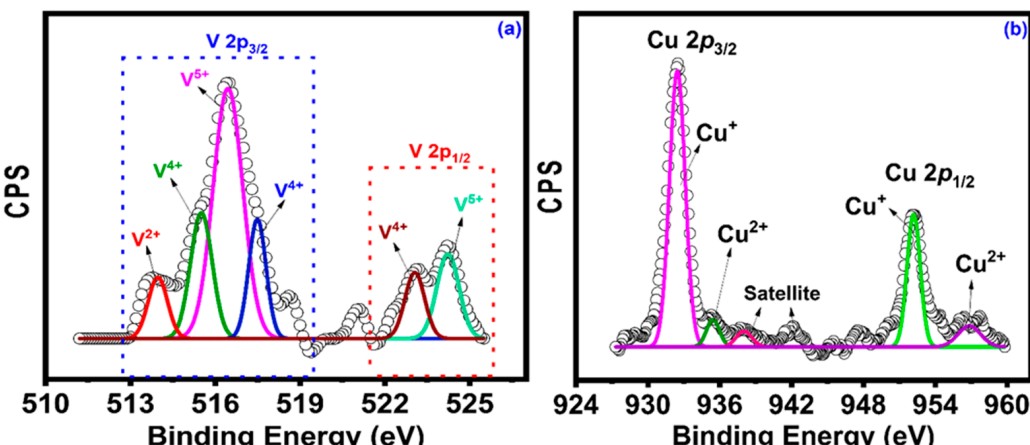

**Figure 7.** (**a**) V 2p core-level XPS spectra and (**b**) Cu 2p core-level XPS spectra.

### 3. Catalytic Degradation of Monocrotophos

The synthesized 0.3 wt% V/Cu-doped gCN has higher photocatalytic mineralization compared with other wt% doped and undoped gCN nanosheets. The excellent photocatalytic ability of thermally decomposed V/Cu/gCN nanosheets is attributed to a higher mineralization ability toward the degradation of MCP (Figure 8a). Owing to its higher photocatalytic ability, V/Cu/gCN supplies a large number of electrons and holes to improve MCP pesticide degradation reactions. Furthermore, the high-dispersion property of the exploited V/Cu/gCN nanosheets in an aqueous solution also results in excellent contact between nanosheets and the MCP pesticide. The higher photocatalytic ability of the V/Cu/gCN nanosheets is due to the direct Z-scheme direct bandgap and the shift in the light-harvesting ability toward the visible range, resulting in the enhancement of the optical bandgap [20,23,25,26]. During the degradation process, oxygen sites appeared on V/Cu/gCN nanosheets with the strong reduction of $O_2$ into superoxide radicals, resulting in extensive photocatalytic mineralization of the target pollutants in the aqueous medium [30]. The degradation efficiency of MCP with the help of V/Cu/gCN (8 mg) nanosheets as a photocatalyst was studied with absorbance changes at the maximum wavelength of the MCP at various time intervals. Therefore, the V/Cu/gCN-nanosheet-assisted MCP degradation efficiency can be obtained using the following mathematical expression [1,2]:

$$\text{Photocatalytic Efficiency (\%)} = (C_t/C_0) * 100 \qquad (2)$$

where $C_t$ and $C_0$ are the MCP absorbance intensity after and before degradation.

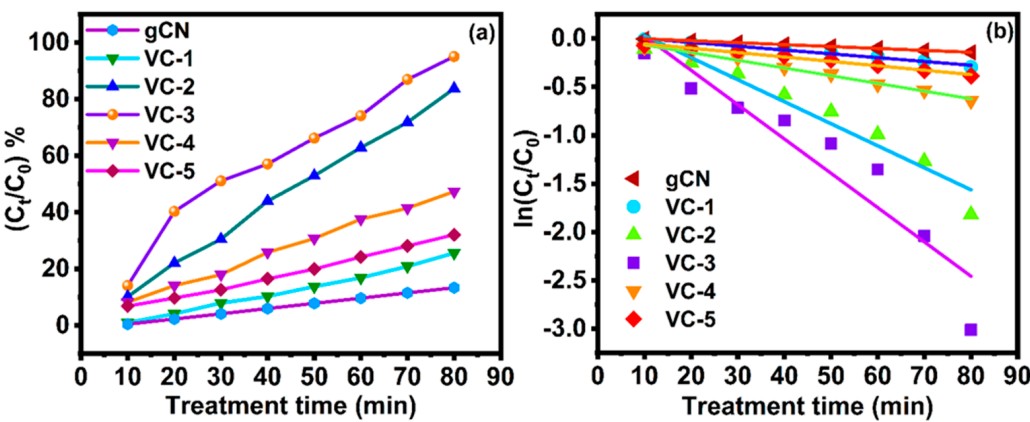

**Figure 8.** MCP degradation efficiency: (**a**) photocatalytic ability of gCN and V/Cu/gCN (VC-1 to VC-5) nanosheets and (**b**) N1s spectra of gCN and V/Cu/gCN nanosheets.

On 0.3 wt% V- and Cu-doped gCN, the MCP pesticide was efficiently mineralized, and a mineralization efficiency of ~95% was obtained in 80 min. On the other hand, the other wt% V/Cu/gCN nanosheets had lower degradation efficiency, which is shown in Figure 8a. The excellent pesticide mineralization was attributed to enormous electron-transfer characteristics and the better light-gathering tendency of V/Cu/gCN nanosheets.

Moreover, the photodegradation process in the presence of V/Cu/gCN nanosheets followed a pseudo-first-order kinetic reaction mechanism for MCP degradation. The linear relationship between MCP degradation and the degradation rate constant, *k*, can be obtained as follows [30]:

$$ln\,(C_t/C_0) = -kt \qquad (3)$$

where *k* is the rate constant of the pseudo-first-order model, and $C_t$ and $C_0$ are the corresponding absorbance intensity values of the MCP solution at time *t* and 0 min, respectively. The linear fit of ln $(C_t/C_0)$ versus treatment time plots (Figure 8b) demonstrated that the degradation of MCP undergoes pseudo-first-order reaction kinetics. The reaction kinetic values are 0.9995, 0.9849, 0.9340, 0.8693, 0.9915, and 0.9875 for gCN, VC-1, VC-2, VC-3, VC-4, and VC-5, respectively.

Figure 9a shows the effect of the photocatalyst concentration on the mineralization of MCP during the treatment time and the degradation ratio ($C_t/C_0$). Figure 9a illustrates the degradation of MCP ($1 \times 10^{-4}$ mole) with catalyst dosages of 2, 4, 6, and 8 mg in 0.3 wt% V/Cu/gCN nanosheets. The amount of V/Cu/gCN was studied for the effective removal of the MCP target pollutant. For this, the nanosheet dosage was varied at 2, 4, 6, and 8 mg, and the degradation reactions proceeded under similar treatment conditions. Among the catalyst dosages, 8 mg exhibited better efficiency, and 95% degradation of MCP was obtained within 80 min under visible-light illumination. However, the MCP removal efficiency was very low with an excessive catalyst dosage, which eventually allowed light radiation to pass into the target pollutant in the solution [23–25,30]. Simultaneously, it induced the formation of various ROS. Figure 9b shows that the prepared nanosheets followed a pseudo-first-order kinetic reaction for MCP degradation. The reaction kinetic values are 0.9832, 0.9794, 0.9732, and 0.8693 for 2, 4, 6, and 8 mg, respectively.

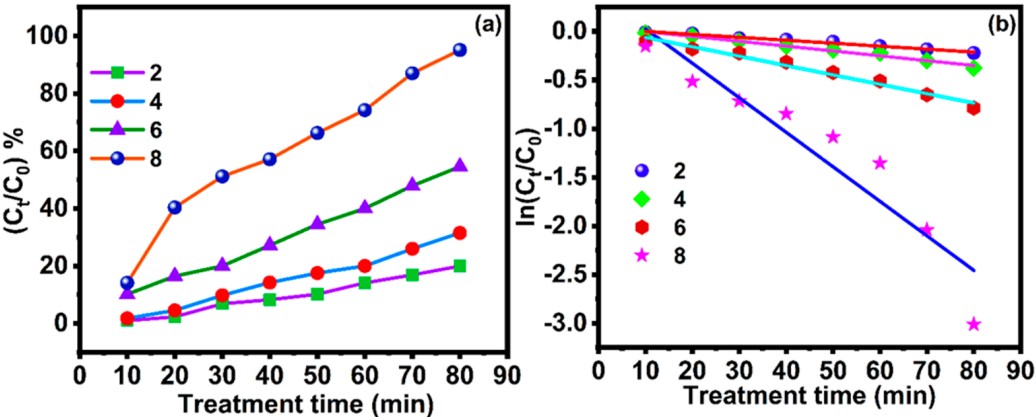

**Figure 9.** Effect of (**a**) photocatalyst concentration on degradation of MCP and (**b**) pseudo-first-order kinetic reaction rate constant.

### 3.1. Effect of pH and Radical Scavengers

The effect of solution pH on the photocatalytic reaction on the surface of the catalyst is the most important because it affects the photocatalyst surface charge characteristics and the size of aggregates it generates. Therefore, the pH role in the photocatalytic mineralization of MCP was observed at pH values of 3, 5, 7, 9, and 11 with a constant MCP concentration and photocatalyst concentration of $1 \times 10^{-4}$ mole and 8 mg. The obtained results are shown in Figure 10a. The pH changes strongly affected the photocatalyst surface and adsorptive interaction of the target pollutants. This is the most important parameter in photocatalyst degradation or oxidation. It is well known that hydroxyl radicals are generated at the photocatalyst surface, and these interact with the positive holes [6–10]. On the other hand, superoxide radicals are formed and strongly interact with the photoinduced electrons. Hence, the electrons serve as reduction sites, which means the generation of more superoxide radicals, and the positive holes function as oxidation sites at pH 7.0 of the reaction medium. Moreover, hydroxyl radicals are the dominant species under neutral conditions. In alkaline and acidic media, reactive species are easily formed near the catalyst surface. Therefore, there exists electrostatic and Coulombic repulsion between the photocatalyst surface and the reactive species. It is noteworthy that the rate of the mineralization of MCP was higher at neutral pH, exhibiting a maximum at pH 7.0, and then the rate of mineralization started decreasing.

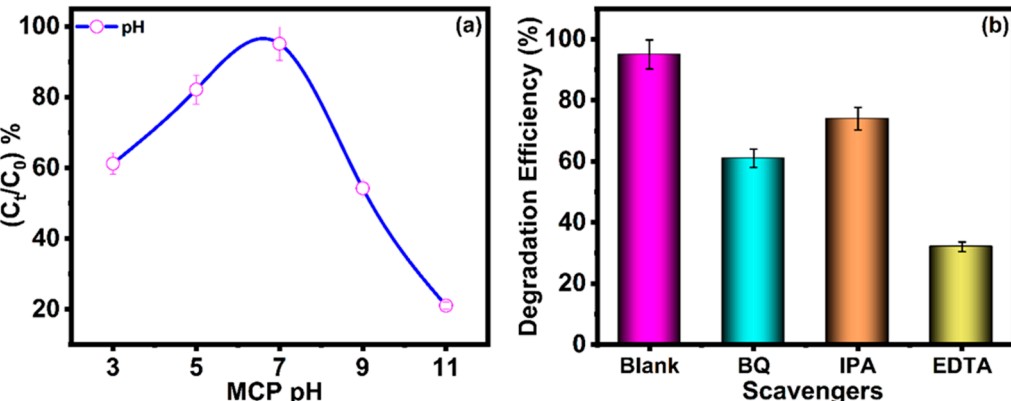

**Figure 10.** (**a**) Effect of pH and (**b**) effect of radical scavengers.

In photocatalytic mineralization, the target effluents are degraded by strong reactive radicals such as superoxide radicals ($O_2^{\bullet-}$), hydroxyl radicals ($^\bullet OH$), and holes ($h^+$). The generation of radicals was observed using a radical-trapping experiment using various radical scavengers (benzoquinone (BQ), isopropyl alcohol (IPA), and EDTA, respectively) [20,30], and the observed results are shown in Figure 10b. In this process, IPA was used to trap OH, and 72% degradation was achieved when compared to the reaction without scavengers (95%). This indicates that OH radicals are not involved in the mineralization process. Moreover, the addition of 1mM EDTA and BQ trapped holes ($h^+$) and superoxide radicals ($O_2^{\bullet-}$). Approximately 30% and 58% MCP mineralization was obtained by EDTA and BQ, respectively. The radical-trapping experiment revealed that superoxide radicals and holes are the most important reactive radicals for the mineralization of MCP. Furthermore, it can be confirmed from the results that superoxide radicals and holes are the major and minor reactive species in the photocatalyst-assisted degradation of MCP by V/Cu/gCN.

*3.2. Photocatalytic Reaction Mechanisms*

The primary mechanism for the operation of semiconductor-based photocatalysts is thought to be the formation of photoinduced electron and hole pairs, as well as migration, separation, and capture by reactive radicals (Figure 11). The light-harvesting efficiency of gCN is enhanced in the visible region due to the incorporation of V and Cu ions [20,23,25,26]. The charge-transfer mechanism during the photocatalytic reaction was investigated by the direct Z-scheme using the edge-band potentials of CuO, $V_2O_5$, and gCN. The conduction band bottom (*CBB*) and valence band top (*VBT*) were studied using the following mathematical expression.

$$VBT = X - E_H - 0.5\,E_g \qquad (4)$$

$$CBB = VBT + E_g \qquad (5)$$

where *X* and *E* are the semiconductor electronegativity and energy possessed by free electrons utilizing the hydrogen scale, respectively. The energy bandgap values were obtained from previously published articles [48–51]. The restriction of photoinduced electron–hole pairs and the generated charge carrier's migration in V/Cu/gCN are extremely effective, and the nanosheet pore structure can effectively reduce the charge carrier transport distance, resulting in high electron mobility. It is well known that, based on the literature and the above-obtained experimental results on the physio-chemical properties, photocatalytic ability, and identified reactive radicals, it is inferred that the photocatalytic ability of V/Cu/gCN is boosted by the incorporation of V and Cu [27,28,30]. In addition, V incorporation significantly enhances the physio-chemical properties of gCN nanosheets and greatly reduces the rate of charge-transfer resistance [52]. Hence, the enhanced properties of V ions in gCN can be attributed to the modified gCN lattice and band structure.

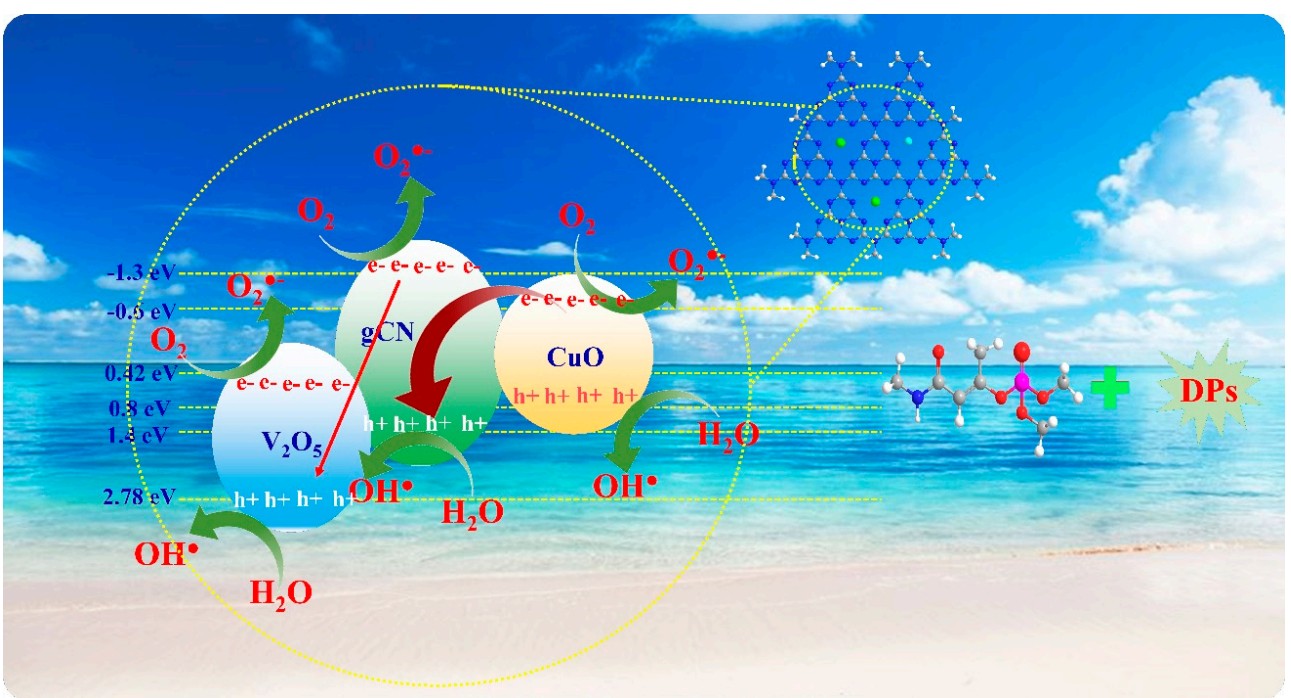

**Figure 11.** Possible photocatalytic reactions for MCP degradation.

As per these results, photoinduced charge pairs are well separated in the doped gCN nanosheets, and doped gCN has a higher interfacial charge moving between the electron acceptor and donor, resulting in a lower rate of recombination. In contrast, the photoinduced holes in the VB of V can directly interact with target pollutants, resulting in the oxidation and degradation of MCP [20,23,25]. Concurrently, the photoinduced electrons can directly transfer to the catalyst surface to interact with the aqueous solution and generate highly reactive radicals such as $OH^\bullet$ and $O_2^{\bullet-}$. These generated reactive species can oxidize the organic effluents due to their higher oxidation potentials. In addition, both $V_2O_5$ and gCN can be initiated by incident light to enhance photoinduced electron–hole pairs. The photoinduced charge-transfer path is like a typical heterojunction system; hence, charged electrons in the CB of $V_2O_5$ will generate fewer superoxide radicals due to its low reducibility. Therefore, the charged electrons in the CB of $V_2O_5$ tend to move and recombine with the charged holes in the VB of gCN. In this way, the larger number of charged electrons transferred in the CB of gCN can reduce $O_2$ into $O_2^{\bullet-}$, which strongly interacts with the pollutant and breaks it down into small molecules.

On the other hand, after visible-light illumination, the formed charge carriers of the electrons in the CB of gCN can be rapidly trapped by Cu ions, generating a Schottky barrier that decreases the recombination of $h_{VB+}$ and $e_{CB-}$. The above classification is evidenced by the PL emission test results. Multivalent V ($V^{4+}$, $V^{5+}$, and $V^{2+}$) and Cu ($Cu^{2+}$ and $Cu^+$) [26,30] coordinated with gCN are involved in the reactive sites to enhance the degradation efficiency of MCP. Nevertheless, at a V and Cu concentration of 0.3 wt%, the decrease in crystalline size, increase in the bandgap, and aggregation of V and Cu may generate a new charge recombination center, which restricts the further recombination of the photocatalyst, resulting in the high photocatalytic ability of V/Cu/gCN. The possible reaction mechanisms are as follows [11–14,20–27,30,47].

$$V/Cu/gCN + h\nu \rightarrow eCB^- + hVB+$$

$$V\ (eCB\text{-}) + O_2 \rightarrow O_2^{\bullet-}$$

$$V^{4+} + H_2O_2 \rightarrow V^{5+} + HO^- + HO^\bullet$$

$$V^{5+} + H_2O_2 \rightarrow V^{4+} + H^+ + HOO$$
$$VO^{2+} + H_2O \rightarrow VO^{2+} + 2H^+ + e^-$$
$$Cu^+ + O_2 \rightarrow Cu^{2+} + O_2^{\bullet-}$$
$$Cu^{2+} + O_2^{\bullet-} \rightarrow Cu^+ + H_2O_2$$
$$Cu^+ + H_2O_2 \rightarrow Cu^{2+} + OH^\bullet + OH^-$$
$$MCP + Reactive\ species\ (OH^\bullet, H_2O_2, O_2^{\bullet-}) \rightarrow Degraded\ pollutants\ (CO_2, H_2O_2, etc.).$$

### 3.3. Possible Degradation Pathway of MCP

The photodegradation of MCP and the identification of the byproduct transformation were further examined by the utilization of GC-MS analysis. GC separated the compounds as per their retention time (RT), which was further confirmed by the mass spectroscopy analysis (Figure 12a). The transferred byproducts' identification was verified either by classifying their fragment particles in the mass spectra or by cross-checking the spectral patterns with the related mass spectra of the National Institute of Standards and Technology library (NIST). The possible MCP degradation mechanism involves its breakdown into non-toxic and smaller compounds, as illustrated in the degradation pathway shown in Figure 12b. Then, the MCP pesticide is degraded into acid and some small non-toxic molecules [2,6]. Moreover, MCP causes the complete removal or partial elimination of dimethyl phosphate, phosphoric acid, acetic acid, or glyoxylic acid. On the other hand, the formation of various reactive radicals via the photogeneration of a catalyst, such as V/Cu/gCN nanosheets, leads to the complete removal of MCP. Therefore, the MCP pesticide was successfully degraded into small fragments in the form of non-toxic molecules with the assistance of photocatalytically active V/Cu/gCN.

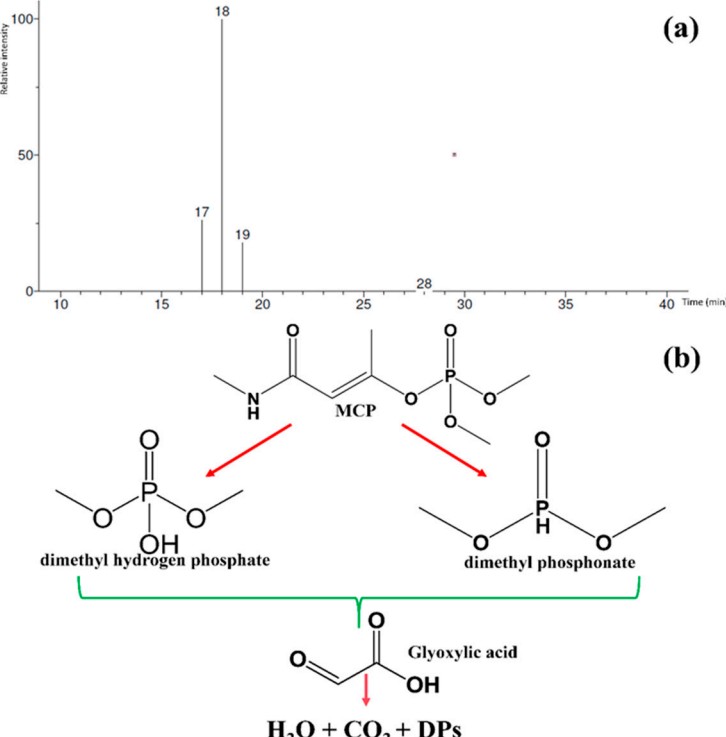

**Figure 12.** (**a**) GC-MS mass spectra of MCP degradation and (**b**) the possible degradation pathways of MCP pesticides.

## 4. Materials and Methods

Urea (CH$_4$N$_2$O, 98%), Ammonium metavanadate (NH$_4$VO$_3$, 99%), Copper Acetate (Cu (CH$_3$COO)$_2$, 98%), Sodium hydroxide (NaOH, 99.30%), Hydrochloric acid (HCl, 37%), and Monocrotophos (C$_7$H$_{14}$NO$_5$P, 36%) were obtained from Sigma Aldrich, Taiwan. All utilized chemicals were analytical grade and were utilized without any further purification processes.

### 4.1. Preparation of V/P-gCN Nanosheets

Different weight ratios of ammonium metavanadate and copper acetate (0.1 to 0.5 wt%) concerning urea were used as source materials of vanadium and copper. Urea was used for the preparation of gCN, and the concentration of urea was fixed at 12 g. At their calculated quantities, these components were dissolved in a water and ethanol solution with a ratio of 2:1. Then, the solution was continuously stirred and heated at 120 °C to obtain a solid product. After that, the solid product was transferred to a 100 mL alumina crucible, which was heated at 550 °C with a ramping rate of 3 °C in a hot-air muffle oven. After thermal decomposition, the reaction product was cooled down naturally with the muffle. Finally, green powder samples were obtained.

### 4.2. Analysis of Prepared Nanosheets

The thermally decomposed V/Cu-gCN nanosheet features were characterized by X-ray diffraction (XRD, D2 Phaser, Bruker, CuK$\alpha$ = 1.540 Å, USA), photoluminescence spectrometry (FP-8300, JASCO), field-emission scanning electron microscopy (FE-SEM/EDX, JEOL, JSM-7610F, and Hitachi Regulus 8100, Tokyo, Japan), and X-ray photoelectron spectroscopy (XPS), (Thermo Scientific Multilab 2000 XPS, Waltham, MA, USA).

### 4.3. Photocatalytic Studies

The photocatalytic features of as-prepared V/Cu-gCN nanosheets were studied using the Monocrotophos pesticide as a model pollutant under visible-light illumination. A tungsten-halogen (150 W, 120 V) lamp was utilized as a source of visible light. The distance between the target pollutant aqueous solution and the light source was fixed at ~10 cm to prevent overheating. In this photocatalytic treatment, $1 \times 10^{-4}$ mole of MCP was dissolved in 100 mL of DI water; before the treatment, the catalyst-added solution was sonicated for half an hour under dark conditions. At every 10 min interval, 5 mL of the treatment solution was collected and filtered using a micro-syringe to eliminate interferences. Finally, the mineralization process was studied using the UV–Visible spectrophotometer (JASCO-750, Tokyo, Japan).

## 5. Conclusions

Vanadium- and copper-doped gCN (V/Cu/gCN) nanosheets were successfully prepared by a thermal decomposition process via the self-assembly of urea and the polymerized products of vanadium and copper precursors. The results of UV-Vis and PL spectroscopy confirmed the improved visible-light absorption with significant charge recombination restriction, increasing the photocatalytic ability of hybrid nanosheets for the removal of pesticide effluents. The increased photocatalytic ability of doped materials in comparison with bulk gCN could be mainly attributed to the relocation of electron–hole pairs through the Z-scheme double-heterojunction charge-transfer mechanism. The photocatalytic removal rate of the optimal V/Cu/gCN photocatalyst (0.3 wt% and 8 mg dosage) for MCP was up to 95% within 80 min under the illumination of visible light. Moreover, the possible photocatalytic reaction mechanisms and possible fragmentation of MCP are also proposed. Therefore, the obtained results confirm the positive effect of V and Cu doping on the photocatalytic ability of gCN and justify its field-scale application for the removal of MCP in water.

**Supplementary Materials:** The following supporting information can be downloaded at: https://www.mdpi.com/article/10.3390/catal12111489/s1, Figure S1. Surface morphologies studies of bulk gCN and V/Cu/gCN nanosheets, (a) bulk gCN (b) VC-1, (c) VC-2, (d) VC-3, (e) VC-4, and (f) VC-5.

**Author Contributions:** Conceptualization, D.V.; methodology, D.V.; validation, S.S., T.-W.C. and L.F.; formal analysis, D.V., C.-L.Y. and Y.-F.Y.; investigation, S.S. and T.-W.C.; characterization, P.-C.C.; writing—original draft preparation, D.V.; writing—review and editing, A.K.K., S.S., C.-L.Y., T.-W.C., L.F. and P.-C.C.; supervision, T.-W.C.; funding acquisition, T.-W.C. All authors have read and agreed to the published version of the manuscript.

**Funding:** This research work was supported by the Ministry of Science and Technology of Taiwan (MOST 109-2221-E-027-059 and 110-2221-E-027-041) and the National Science and Technology Council of Taiwan (NSTC 111-2221-E-027-104).

**Data Availability Statement:** Data available on request due to privacy/ethical restrictions.

**Acknowledgments:** The authors are grateful to the Precision Research and Analysis Centre of the National Taipei University of Technology for providing the instrument facilities.

**Conflicts of Interest:** The authors declare no competing interests.

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
