# Peer review of "Visible-Light-Active Vanadium and Copper Co-Doped gCN Nanosheets with Double Direct Z-Scheme Heterojunctions for Photocatalytic Removal of Monocrotophos Pesticide in Water"

_catalysts, doi:10.3390/catal12111489_

Round 1

Reviewer 1 Report

In this manuscript, the authors reported A novel visible light active Vanadium and Copper co-doped 2 gCN nanosheets with double direct Z-scheme heterojunctions 3 for photocatalytic removal of monocrotophos pesticide in water. In my opinion, this manuscript is interesting to the readers. The topic is relevant in this field. However, there are several issues needed to be improved and revised before the possible publication in catalysts.

1)      The authors main focus lies on formation of Z-scheme but in introduction there is not sufficient information about Z-scheme. Include the recent papers Journal of Alloys and Compounds 898 (2022) 162779, Environmental Research 215 (2022) 114140, Advanced Powder Technology 32 (2021) 3770–3787 and revise the introduction portion.

2)      What is the relation between bandgap and crystallite size? The bandgap fluctuates while crystallite size gradually decreases with doping.

3)      The authors have determined the particle size from TEM/SEM. Why is it important to determine the crystallite size? I could not find any discussion comparing the results with the crystallite size obtained from XRD analysis. Which technique provides a more accurate value for the average particle size?

4)      Calculate the particle size from TEM and compare it with XRD results following the article Advanced Powder Technology 33 (2022) 103708.

5)      The authors reported that VC-3 has minimum bandgap, while in VC-5, both elements were doped. What is the reason for reduction in bandgap for VC-3?

6)      PL results are not clear in diagram. Revise the PL figure and explain the results.

7)      What is the correlation between the PL and XPS results? Justify your answer.

8)      In Photocatalytic mechanism, it is better to draw energy band diagram and explain the dopant effect and explain mechanism. The author may take help from the Surfaces and Interfaces 34 (2022) 102376.

9)      Prepare a comparision table for the degradation efficiency against targeted material using gCN based materials.

10)  What is the novelty of the current findings?

Author Response

Thanks for the reviewer's suggestion, as per the reviewer's suggestion we have corrected and improved this revised manuscript. This current version of the revised manuscript is suitable for publication.

Reviewer 2 Report

The manuscript catalysts-1977108 describes gCN nanosheet photocatalysts co-doped with Cu/V for photocatalytic removal of monocrotophos in an aqueous solution. The manuscript has several language errors and is difficult to read and understand. Some discussions are not supported by scientific pieces of evidence. Characterizations must be revised. I do not recommend the publication of this manuscript in its present form.

Some main points addressed to the authors can be seen below.

11)      Some reference citations are not properly formatted (lines 43, 52, etc.).

22)      Please revise the sentence “cause to can cancer” – line 54.

33)      Please revise the sentence “preventing the insects from toxic the cotton plants” – Line 57.

44)      Please revise lines 70 – 71.

55)      Add a reference to Line 80.

66)      Revise Line 84 “there are no reports are available for”.

77)      I could not follow the idea of “direct Z-scheme for two heterojunction nanosheets” – Line 87. Authors must revise the idea.

88)      Figure 1 is ahead of the text in the Results and Discussion section. It must be corrected.

99)      As the samples have less than 0.5 wt% of Cu/V, any expected diffraction peaks from Cu/V oxides, or related compounds, are not expected to appear since the limitations of the technique (lines 108 – 111). Raman spectra are necessary to complement the structural properties.

110)  How were micro-strain and crystallite sizes calculated in Table 1?

111)  Line 99 is incomprehensible.

112)  Line 103. The mentioned shift is not clear in Figure 1. Additionally, why does the intensity of the 002 peak decrease so much?

113)  Line 115 is incomprehensible.

114)  The straight lines used for bandgap calculation (Figure 2) are wrong. The author must revise all optical characterization before resubmitting the manuscript.

115)  Line 152. The emission PL intensity is not “very low”. There is no peak there and therefore the electronic structure must be completely different. Authors must revise the data and analysis. PL's assumption of defect sites is speculative. The authors must provide scientific evidence.

116)  I cannot see a sheet-like structure in Figure 4a-f (lines 172 and 173). The morphological features must be better discussed.

117)  Line 175. You cannot study topography with an HRTEM microscope.

118)  Figure 5. Authors must present a legend identifying figures (a) and (b).

119)  Lines 247 and 248 should come after the description of the results.

220)  Error bars should be presented in Figure 8. Were the experiments reproduced in triplicates?

221)  Section 3 is very confusing as presented.

222)  VC-3 and VC-2 sampled do not follow a pseudo-first-order kinetic. Same for 8 mg in Figure 9.

223)   Why are 0 min data points not presented in Figures 8 and 9?

224)  Lines 308 – 309. It is impossible to follow the sentence “Hence, the negative electrons serve as reduction sites and the positive holes imitated the oxidation sites”.

225)  Columbia repulsion? Line 312.

226)  What is the pH in Figure 10b?

227)  Photocatalytic Reaction Mechanisms must be completely revised. Several errors are present. Examples (not limited to): (i) V incorporation significantly enhances the conductivity of gCN nanosheets (Line 344)?; and (ii) the discussion from lines 346 – 348 is completely speculative.

228)  The purity of reagents was not provided.

229)  Incident light intensity (mW/cm2) must be provided.

330)  Line 432 – “lamb”.

331)  GC-MS experimental conditions must be presented.

332)  From where do the authors state for a direct Z-scheme heterojunction? There is no heterojunction!!!

Author Response

(The authors gave the same response as above.)

Round 2

Reviewer 2 Report

Although the authors have resubmitted the manuscript, its content still contains serious scientific errors and interpretations. The characterization of materials needs to improve. XPS is not a structural characterization, therefore measurements and Raman analysis are necessary. The non-identification of V and Cu oxide phases in XRD is not a guarantee of successful doping. As stated in the first review, this is only due to the technique's detection limit.

The identification of oxidized V/Cu on the surface of the material is not enough for Z-scheme.

To determine the bandgap authors should not use a tangent line as done in Figure 2 but should use a line segment of the absorption curve close to the edge threshold. This needs to be fixed for future submissions.

In the revised version the authors included the creation of intragap states, but the discussion is speculative and lacks scientific evidence.

In line 190, the HRTEM technique does not present enough statistics to guarantee the effective incorporation of V/Cu.

Were the photocatalytic experiments repeated? Please display the error bars in the data.

There is a serious conceptual error in the manuscript. All the time the authors stand about the incorporation of Cu/V as dopants, but in the end, they defend a Z-scheme heterojunction. Results are inconsistent.

Title is too long and needs to be rethought.

Author Response

Thanks for the reviewer's suggestions and comments, as per the reviewer's comments we have rectified all the corrections and the revised version of our manuscript is more suitable for publication.

Thanking You
